# A Multiscale Neighbor-Aware Attention Network for Collaborative Filtering

Jianxing Zheng [1], Tengyue Jing [2], Feng Cao [3,*], Yonghong Kang [3], Qian Chen [3] and Yanhong Li [3]

[1] Institute of Intelligent Information Processing, Shanxi University, Taiyuan 030006, China; jxzheng@sxu.edu.cn
[2] North Automatic Control Technology Institute, Taiyuan 030006, China; 202022408027@email.sxu.edu.cn
[3] School of Computer and Information Technology, Shanxi University, Taiyuan 030006, China; 202322405009@email.sxu.edu.cn (Y.K.); chenqian@sxu.edu.cn (Q.C.); liyh@sxu.edu.cn (Y.L.)
* Correspondence: caof@sxu.edu.cn

**Abstract:** Most recommender systems rely on user and item attributes or their interaction records to find similar neighbors for collaborative filtering. Existing methods focus on developing collaborative signals from only one type of neighbors and ignore the unique contributions of different types of neighbor views. This paper proposes a multiscale neighbor-aware attention network for collaborative filtering (MSNAN). First, attribute-view neighbor embedding is modeled to extract the features of different types of neighbors with co-occurrence attributes, and interaction-view neighbor embedding is leveraged to describe the fine-grained neighborhood behaviors of ratings. Then, a matched attention network is used to identify different contributions of multiscale neighbors and capture multiple types of collaborative signals for overcoming sparse recommendations. Finally, we make the rating prediction by a joint learning of multi-task loss and verify the positive effect of the proposed MSNAN on three datasets. Compared with traditional methods, the experimental results of the proposed MSNAN not only improve the accuracy in MAE and RMSE indexes, but also solve the problem of poor performance for recommendation in sparse data scenarios.

**Keywords:** multiscale neighbors; attentional mechanism; collaborative embedding; recommendation

## 1. Introduction

Collaborative filtering has become a fundamental technology in e-commerce platforms, which has made remarkable achievements. Collaborative filtering assumes that similar users have similar interests, and makes recommendation services for the target user based on similar users' interests. In most e-commerce scenarios, the interaction between users and items reflects users' interest preferences, which is often used to find similar neighbors to model collaborative representations for users and items [1,2].

However, the e-commerce platform generates a large number of new users and product items every day. Some users have not rated or purchased new products, which leads to fewer interactive records. Thus, it is difficult to learn high-quality embedding representations for users and items. As a result, collaborative filtering still faces the problem of interaction data sparsity in recommender systems [3]. Aiming to solve the sparse recommendation problem, most recommender systems leverage auxiliary attribute information of users and items to establish users' preferences and mine similar neighbors for collaborative filtering [4]. However, similar neighbors mainly come from an attribute preference of uniform scale. In fact, different users have various social attributes, and different items have multiple product descriptions. Different combinations of attributes are conducive to generating various types of neighbors to describe semantic characteristics of different granularity for nodes. That is, when some users have multiple attributes and the other users have a few attributes, we can make collaborative recommendations according to their multiscale attribute neighbors. In addition, the interaction rating of users reflects their differentiated preference for items. We can leverage differentiated rating preference via

different rating neighbors to explore the fine-grained preference motivation. Thus, how to deal with collaborative filtering by combining multiscale attribute neighbors with rating neighbors is an important task. Multiscale node embedding describes the fine-grained semantic representations from multiple perspectives, and can effectively solve the problem of poor performance of sparse recommendation, which is of great significance for industrial applications.

In the e-commerce recommender systems, multiscale attribute combinations can produce single-attribute-view neighbors and multi-attribute-view neighbors. For example, in movie recommender systems, users with the same gender and age have more similar interest behaviors than users of the same gender. As a result, different types of attribute-view neighbors can be constructed in various attribute combination spaces. In addition, different interaction-view neighbors can be obtained according to the types of interaction ratings, such as 1–5. We model the interaction-view neighbor embedding of nodes on various interactive views. The attention mechanism [5] is used to focus on specific input features, analyze the importance of all aspects of input features, and improve the expression ability of the model, which has been widely applied in the fields of natural language processing and image processing. Inspired by [6], this paper captures a different type of attribute neighbor embedding and interactive neighbor embedding and mine their collaborative signals with the attention mechanism to model multiscale node embedding. The multiscale node embedding can effectively capture diverse semantics of nodes from different types of neighbors to enhance sparse recommendation.

To summarize, the main contributions of this paper are as follows.

- Various neighbor graphs of attribute tag and rating tag are designed to learn attribute neighbor embedding and interaction neighbor embedding, which capture embedding signals of various neighbors at different levels.
- An attention network is developed to refine the collaborative semantics of multiscale neighbors, which is utilized to filter the irrelevance signals of various types of neighbors.
- A joint learning of multiscale neighbor embedding is proposed for rating prediction, which solves the problem of poor accuracy in the context of sparse recommendation.

The rest of this paper is structured as follows. Section 2 outlines related work, including the state-of-the-art of neighbor-based recommendation and attention mechanism. In Section 3, we present the framework of the proposed multiscale neighbor-aware attention network. Section 4 provides the methodology of the MSNAN recommendation. Section 5 describes the experimental setup and evaluation. Section 6 gives the experimental results and analysis. Finally, Section 7 concludes this work.

## 2. Related Work

In this section, the related work on neighbor-based recommendations of collaborative filtering and attention mechanisms are briefly reviewed.

### 2.1. Traditional Collaborative Filtering

Traditional collaborative filtering utilizes the nearest neighbors of a user or an item to generate recommendation results. Most popular methods leverage matrix factorization to learn the latent factors of users and items in terms of their historical information, such as BiasedMF [7], PMF [8], SVD++ [9], and LLORMA [10]. The decomposed low-dimensional factor vectors can be used to predict the user's rating preference for items. These methods mainly depend on the rating interactions between the user and item and have low performance in the case of sparse data. Thus, some side information is merged into matrix factorization to alleviate the sparsity of interaction records [11,12]. SSLIM [13] develops a sparse aggregation coefficient matrix by considering the user–item profiles and side information of items. The auto-encoder and -decoder techniques are used to learn the latent factors of nodes for collaborative filtering [14]. Park et al. [15] developed a group recommender system to select the suited recommendation items for store product

placement. In recent years, some matrix decomposition models of deep learning have been studied [16,17]. DeepFM learns low-order and high-order interaction features of compressed interaction neighbors through a neural network [18]. Cai et al. [19] leveraged various multi-grained sentiment features and latent factors of matrix factorization to obtain sufficient representations of users and items to make rating predictions. Although these models have the ability to handle the sparsity recommendation problem, they leverage raw neighbors to learn their high-order interaction features and have limitations on processing different contributions of fine-grained neighbors.

### 2.2. Deep Learning-Based Collaborative Filtering

Deep learning-based methods can utilize the interaction neighbors between users and items to extract their latent vectors, which achieves excellent performance. Deep neural networks take advantage of both side information and feedback information to model linear and nonlinear features. For example, NFM [20] encodes interaction neighbor IDs of users, items, and their features into different vectors. Wide and Deep learning [21] combines the linear model and the deep neural network to implement efficient recommendations for scenarios with sparse data. Some works transfer diverse interactions of neighbors to learn rich features of nodes [1]. MCCF decomposes and recombines the latent components of user–item interaction graph to capture fine-grained user preference [22]. Aiming to solve the cold-start problem, Magron et al. [23] considered content information of acoustic features to learn the interaction between users and songs and proposed a neural content-aware collaborative filtering framework for music recommendation. Graph neural networks can learn the interaction characteristics of multi-order neighbors. MBGCN leverages multiple types of user-to-item interactions and the similarity item-to-item to propagate neighbor semantics [24]. Multiple neighbors on the path are used to capture the diversity of user interests and improve the accuracy of personalized recommendations [25]. User–item neighbor interaction and item–item neighbor relevance were leveraged to model a two-hop paths-based deep network to improve user engagement [26]. Tai et al. [27] designed a user-centric two-level path network in terms of entities of knowledge graph to generate user portfolio information. Duan et al. [28] learned the features of nodes from different dimensions of time and position and investigated the consistency of two representations for sequential recommendation. Most of neighbor embedding algorithms exploit interactive neighbors and do not distinguish the representations of different types of neighbors.

### 2.3. Attention Network for Collaborative Filtering

An attention mechanism is applied in collaborative filtering recommendations by identifying different importances of neighbors [5,29]. DGCF identifies the importance of diverse user–item interaction neighbors and models fine-grained intent-aware graph collaborative filtering [30]. By integrating content-based with collaborative filtering, ACCM considers the importance of end-to-end features and traditional attributes adaptively via an attention mechanism to handle the cold-start problem [3]. Chen et al. designed both item-level and component-level attention networks for multimedia recommendation [31]. Different auxiliary information can be learned to enhance important signals for neural networks. By leveraging intra-entity interaction and inter-entity interaction, AKUPM explores the relationships between users and other entity neighbors for alleviating the sparsity recommendation problem [32]. The attentional mechanism can provide different functions of multi-type entities on the knowledge graph and help obtain important neighbor information on different paths. KGAT learns the importance of higher-order neighbors of nodes in knowledge graphs by considering various features of entities [33]. A knowledge-aware attention mechanism is adopted to discriminate the contributions of different collaborative neighbors for recommender systems [34]. There are considerable advantages to applying the attention model in graph-based neural networks. A neural co-attention model utilizes auxiliary information of meta-based neighbors for top-n recommendation of heterogeneous information networks [35]. By leveraging the higher-order friends in the social network,

Xiao et al. [36] designed a social explorative attention network to make personal interest recommendations. Ye et al. [37] utilized both influence graph and the preference graph to fuse different user and item embeddings to make rating predictions. However, most attentional models deal with the role of the same type of neighbors, which limits the discriminative contributions of various neighbors to the user's overall decision making.

## 3. Framework of Multiscale Neighbor-Aware Attention Network

Figure 1 shows the framework of the proposed multiscale neighbor-aware attention network. The framework comprises four components: (1) attribute-view neighbor node embedding, (2) interaction-view neighbor node embedding, (3) attentional multiscale neighbor node embedding, and (4) rating prediction. In the framework, attribute-view neighbor node embedding is used to learn the different roles that similar neighbors with different attribute sets play in collaborative filtering. The interaction-view neighbor node embedding learns the role of similar neighbors with the same rating behavior in collaborative filtering. Neighbors with different attribute sets can form multiscale similar neighbors. Considering that users with different attributes have different rating behavior habits, we model the preferences of attribute sets for rating behavior through a multiscale neighbor-aware attention network. The attention network measures the collaborative contribution of coarse-scale attribute neighbors and fine-scale attribute neighbors to the rating decisions of target users.

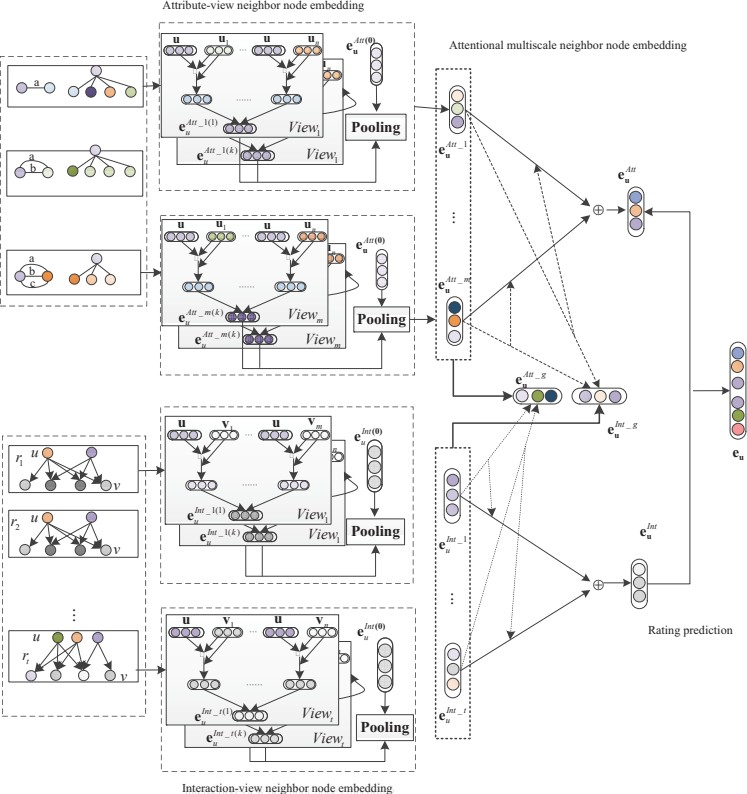

**Figure 1.** The framework of the multiscale neighbor-aware attention network.

In a nutshell, the framework works as follows. For the attribute-view neighbor node embedding, we first construct an attribute-view neighbor graph according to the association of a node on an attribute set such as {a} or {a,b}. Attribute sets of different scales induce multiple types of attribute neighbors. Based on various attribute-view neighbors, graph neural networks are used to obtain attribute-view neighbor node embedding.

For the interaction-view neighbor node embedding, we divide different rating tag spaces according to different rating grades. Under different rating tag spaces, we form similar neighbors with different rating behaviors. Then, graph neural networks are lever-

aged on different user–item interaction neighbor graphs to obtain various interaction-view neighbor node embedding.

Then, an attention mechanism is utilized to estimate the interactive contributions between various attribute neighbors and interaction neighbors. For the attribute-view and interaction-view neighbors, we calculate the global attribute neighbor collaborative signal and interactive neighbor collaborative signal, respectively. Meanwhile, we match the local collaborative signals between different-scale attribute neighbors and interactive neighbors. Considering the global and local semantic information provided by different neighbors, multiscale neighbor node embedding is computed to capture rich collaborative signals.

Finally, based on multiscale user embedding and item embedding, the inner product can be used to predict the rating score of the user to the item.

## 4. Methodology
### 4.1. Attribute-View Neighbor Embedding

In e-commerce networks, some attribute descriptions describe characteristics of users or products, which are helpful to discover various types of similar users or similar items. In this subsection, we utilize various types of attribute sets to calculate similar neighbors of different scales. Then, we learn the nodes' attribute-view neighbor embedding in terms of different-scale neighbor graphs.

Usually, users have various kinds of attributes, such as gender, age, occupation, and so on, which reflects users' interest preference to a certain extent. For example, users with the same gender can form a neighbor graph with a coarse-grained perspective, while users with the same gender and age can build a neighbor graph with a fine-grained attribute space. A coarse-grained neighbor can provide robust interest preferences for cold-start recommendation. A fine-grained neighbor helps discover refined similar preferences and model accurate collaborative recommendation. Based on this assumption, we can construct different views of attribute neighbor graphs to incorporate signals of various neighbors for modeling the embedding of nodes.

Given an attribute $a$, we can define a neighbor set $N_u^a$ of user $u$ on an attribute $a$ as follows:

$$N_u^a = \{u' | f_a(u) = f_a(u')\} \tag{1}$$

where $f_a(u)$ is the attribute value of the user $u$ on attribute $a$. $N_u^a$ describes the collaborative neighbors with the same attributes as user $u$. Considering all attribute value types in the set $A$, we can construct the user-attribute relation matrix as $\mathbf{M}_{U \times A}$. Then, based on the user distribution of the set $A$, we can establish the neighbor relationship matrix of users by $\mathbf{MM}^T$, labeled as $\mathbf{M}_{U \times U}$. Here, various attribute-view neighbors reflect multiscale collaborative preference, which can affect the decision-making tendency of the target user.

Based on the idea in [33,38], we can define the collaborative signal of a first-order neighbor $u'$ for user $u$ on the attribute $a$ as follows.

$$\mathbf{m}_{u \leftarrow u'}^a = \frac{\mathbf{u} + (\mathbf{u} \odot \mathbf{u'})}{|N_u^a||N_{u'}^a|} \tag{2}$$

Here, $\mathbf{m}_{u \leftarrow u'}^a$ represents the influence of similar neighbors in terms of attribute $a$ on the target user $u$. $\mathbf{u}$ is the initialized embedding vector. Thus, considering all the first-order neighbors, attribute $a$-view collaborative signal for user $u$ can be defined as $\mathbf{e}_u^{a(1)} = \sum_{u' \in N_u^a} \mathbf{m}_{u \leftarrow u'}^{a(1)}$. As is known, neighbors with the same attributes tend to spread their preferences in the social network. Considering the spread contributions of $k-1$ hop neighbors, the recursive collaborative signal of neighbor $u'$ for user $u$ on the attribute $a$ can be formulated as follows [33,38].

$$\mathbf{m}_{u \leftarrow u'}^{a(k)} = \frac{\mathbf{u}^{(k-1)} + (\mathbf{u}^{(k-1)} \odot \mathbf{u'}^{(k-1)})}{|N_u^a||N_{u'}^a|} \tag{3}$$

Further, attribute $a$-view recursive collaborative signal for user $u$ can be defined as $\mathbf{e}_u^{a(k)} = \sum\limits_{u' \in N_u^a} \mathbf{m}_{u \leftarrow u'}^{a(k)}$. We adopt the average pooling to obtain the attribute $a$-view neighbor-aware node embedding for user $u$ as $\mathbf{e}_u^{Att\_a} = agg(\mathbf{e}_u^{a(1)}, \cdots, \mathbf{e}_u^{a(k)})$. Here, some aggregation strategies can be used to fuse different orders of neighbor embedding.

Different types of attributes can induce different similar neighbors. Considering various types of neighbors in other attribute views, we can obtain attribute $A$-view neighbor-aware user embedding as $\mathbf{e}_u^{Att\_A} = avg(\mathbf{e}_u^{A(1)}, \cdots, \mathbf{e}_u^{A(k)})$. Similarly, given an item $v$, we adopt different types of item neighbors to obtain the attribute-view neighbor-aware item embedding.

Different users have special behavioral perceptions of rating labels, which can be used to explore users' behavioral preferences in fine granularity. Thus, according to the types of rating labels, we also divide different interaction view spaces with rating labels and construct various rating interaction graphs for learning user and item embeddings. For example, we can think of item groups with the same rating as neighbors of a user with the same scale preference. Based on different rating labels, we model interaction-view neighbor-aware embedding with different rating neighbors, which can be defined as $\{\mathbf{e}_u^{Int_1}, \cdots, \mathbf{e}_u^{Int_r}\}$. Then, interaction-view neighbor-aware item embedding of an item $v$ can be defined as $\{\mathbf{e}_v^{Int_1}, \cdots, \mathbf{e}_v^{Int_r}\}$.

### 4.2. Cross Attention-Based Multiscale Neighbor Embedding

In the homogeneous attribute view, the multiscale neighbor-aware embedding signals with different granularity can provide diversified collaborative signals for node representation. In addition, various neighbor-aware embeddings of attribute view and interactive view can be used to model the heterogeneous collaborative signals, which can enrich and enhance the representation ability of node embedding.

According to neighbor-aware node embedding on different attribute-view spaces, we can model global attribute neighbor-aware node embedding for user $u$ as follows.

$$q_i = \mathbf{h}_i \mathbf{e}_u^{Att\_i} \tag{4}$$

Equation (4) describes the influence of user embedding in the attribute $i$ space on the global attribute neighbor-aware user embedding. $\mathbf{h}_i$ is the parameter vector. Considering different user embeddings of $m$ spatial types, the normalized weights are defined in Equation (5).

$$\alpha_i = \frac{e^{q_i}}{\sum\limits_{s \in \{1,...,m\}} e^{q_s}} \tag{5}$$

Then, the global attribute neighbor-aware node embedding for user $u$ is defined as follows.

$$\mathbf{e}_u^{Att\_g} = \sum\limits_{i \in \{1,...,m\}} \alpha_i \mathbf{e}_u^{Att\_i} \tag{6}$$

In Equation (5), different neighbor-aware user embedding representations depict discriminative semantic information from various attribute-induced neighbors, which provides collaborative signals of neighbors comprehensively for target user's decisions. The global collaborative embedding $\mathbf{e}_u^{Att\_g}$ of various neighbors can improve the semantic ability for recommender systems. Similarly, we can obtain the global collaborative embedding of interaction view spaces as $\mathbf{e}_u^{Int\_g}$.

Given the item $v$, we can also obtain global attribute-neighbor-aware item embedding and interaction-neighbor-aware item embedding as $\mathbf{e}_v^{Att\_g}$ and $\mathbf{e}_v^{Int\_g}$.

Further, taking into account the collaborative signals matched by different neighbor embedding from the attribute and interactive views, we utilize cross-attention to model matched neighbor embedding. Based on the global collaborative embedding of interaction

view, we can compute matched neighbor embedding of attribute view for user $u$ as follows.

$$\beta_i = \mathbf{e}_u^{Int\_g} \mathbf{e}_u^{Att\_i} \tag{7}$$

Here, Equation (7) defines the preferential influence of user embedding in attribute $i$'s view on the user's rating decision. Then, we normalize this influence using Equation (8).

$$\gamma_i = \frac{e^{\beta_i}}{\sum\limits_{s \in \{1,\dots,m\}} e^{\beta_s}} \tag{8}$$

The normalized weight $\gamma_i$ reflects the influence of multiscale attribute neighbor embedding for the user's interaction rating. Furthermore, the attribute embedding by incorporating user rating behavior preference can be updated as in Equation (9).

$$\mathbf{e}_u^{Att} = \sum\limits_{i \in \{1,\dots,m\}} \gamma_i \mathbf{e}_u^{Att\_i} \tag{9}$$

The multi-type matched signals based on attribute neighbors and interactive neighbors can enhance the embedding representation of nodes. Similarly, we can calculate the matched neighbor embedding of the interactive view as below.

$$w_j = \mathbf{e}_u^{Att\_g} \mathbf{e}_u^{Int\_j} \tag{10}$$

Here, $w_j$ is the dependency influence of user embedding in rating tag $j$'s view on user's attributes. In terms of different rating levels, this dependence effect can be normalized using Equation (11).

$$g_j = \frac{e^{w_j}}{\sum\limits_{p \in \{1,\dots,t\}} e^{w_p}} \tag{11}$$

$$\mathbf{e}_u^{Int} = \sum\limits_{p \in \{1,\dots,t\}} g_j \mathbf{e}_u^{Int\_j} \tag{12}$$

In Equation (12), $\mathbf{e}_u^{Int}$ represents the user's interaction embedding incorporating the dependency of user attributes. Considering the matched neighbor embedding in the attribute view and neighbor embedding in the interactive view, we can model fused multiscale neighbor embedding with the concatenation operator as follows.

$$\mathbf{e}_u = \mathbf{e}_u^{Att} || \mathbf{e}_u^{Int} \tag{13}$$

*4.3. Rating Prediction*

Based on the fused multiscale neighbor embedding of users and items, the dot product is used to predict the rating of the user $u$ on an item $v$, which is shown as follows:

$$\hat{y} = \mathbf{e}_u \cdot \mathbf{e}_v + b_g + b_u + b_v \tag{14}$$

Here, the parameters $b_g$, $b_u$, and $b_v$ are global bias, user bias, and item bias. To preserve the preference information of user attributes over item attributes, we define rating prediction of user $u$ for the item $v$ with their attribute-view neighbor embeddings as follows:

$$\hat{y}_{Att} = \mathbf{e}_u^{Att} \cdot \mathbf{e}_v^{Att} + b_g + b_u + b_v \tag{15}$$

Similarly, we can compute the rating with the interaction-view neighbor embeddings in interactive space, which is shown as below.

$$\hat{y}_{Int} = \mathbf{e}_u^{Int} \cdot \mathbf{e}_v^{Int} + b_g + b_u + b_v \tag{16}$$

In the process of model optimization, to observe the influence of attribute neighbors and interaction neighbors on rating prediction, we define the joint root mean squared error (RMSE) loss function as follows.

$$L = \lambda RMSE(\hat{y}, y) + \lambda_{Att} RMSE(\hat{y}_{Att}, y) + \lambda_{Int} RMSE(\hat{y}_{Int}, y) \tag{17}$$

The joint loss function considers the global and local important neighbors to predict the user's rating of the item, which not only captures the user's attribute preference for the item from various attribute-view neighbors, but also retains the behavioral preference of collaborative fine-grained interaction neighbors.

## 5. Experiments

In this section, we verify the performance of the proposed model with the aim of answering the following three questions:

- RQ1: How does MSNAN perform compared with state-of-the-art neighbor-based collaborative filtering methods?
- RQ2: How does the multiscale neighbor node embedding perform for sparsity recommendation?
- RQ3: How do different types of neighbor embedding affect the performance of the model?

### 5.1. Dataset

We ran the proposed MSNAN model and baselines on three public datasets: Movielens-100kr (ML-100kr) (https://grouplens.org/datasets/movielens/ (accessed on 24 September 2023)), Book-Crossing-10croe (BK-10C) (http://bookcrossing.com (accessed on 24 September 2023)), and Douban (https://movie.douban.com/ (accessed on 24 September 2023)) datasets to verify their effectiveness. The ML-100kr contains the interaction ratings of 943 users on 1682 movies. Users have 5 attributes, and movies have 19 attributes. The rating score is on the scale 1–5. For the BK-10C dataset, we selected users who had rated at least 10 books and books that have been rated by at least 10 users, which involved 1820 users and 2030 books. The rating score uses the range of 1–10. The Douban dataset contains ratings of 6971 movies from 3022 users with rating values of 1–5 [39]. On all datasets, the higher the rating is, the more the user likes the movie/book. Statistical information of experimental datasets is shown in Table 1. During the experiment, we use the MAE and RMSE indexes for performance evaluation. The smaller MAE and RMSE values indicate better performance. All the datasets were divided into training set, validation set, and testing set with a proportion of 8:1:1. The rating prediction performance of the model is evaluated on the testing set.

**Table 1.** Statistical information of experimental datasets.

| Datasets | Users | Items | Interactions | Rating | Sparsity |
|---|---|---|---|---|---|
| ML-100kr | 943 | 1682 | 100,000 | 1–5 | 94.12% |
| BK-10C | 1820 | 2030 | 41,456 | 1–10 | 98.87% |
| Douban | 3022 | 6971 | 195,493 | 1–5 | 99.07% |

### 5.2. Baseline Models

To test the contribution of MSNAN method, we employ traditional methods, deep learning-based recommendations, and GCN-based models to compare the performance. Several baselines are as follows.

(1) NCF [17]. A neural network method is used to learn the interaction information between users and items for collaborative filtering.

(2) Wide and Deep [20]. A method combining a generalized linear model and deep neural network is designed to improve the performance of recommender systems.

(3)   NGCF [33]. A collaborative filtering method based on a graph neural network learns the embedding representations of users and items with a user–item interaction graph.

(4)   GCN [40]. A graph convolutional neural network leverages the information of multi-order neighbors by superimposing several convolutional layers for recommendation.

(5)   LightGCN [38]. The embedding representations of user and item are learned by aggregating linear neighbor information of nodes.

(6)   GAT [41]. A graph convolutional neural network method together with attention mechanism learns weighted node embedding representation for recommendation.

(7)   ACCM [3]. The attention mechanism is used to integrate a content-based method with collaborative filtering for rating prediction.

(8)   AFM [5]. An attentional network factorization method learns the interactive importance of different features for prediction.

(9)   TANP [42] A task-adaptive neural network is constructed to learn the relevance of different tasks for user cold-start recommendations.

### 5.3. Parameter Settings

In the experiment, we adopt a grid search to obtain the parameters for optimizing the performance of model. For the attention mechanism, we tune its dimension with the values {32, 64, 128, 256}. To prevent the model overfitting, we adjust the dropout values {0.1, 0.2, 0.3, 0.4, 0.5}. We utilize stochastic gradient descent to optimize the model with a learning rate of 0.01. In order to obtain a better comparison, we use inner product in the prediction layer for NGCF, GCN, LightGCN, and GAT methods to make rating predictions.

### 5.4. Experiment Settings

For the proposed multiscale neighbor method, the NGCF graph convolution model is used in Equations (2) and (3) to capture the cooperative signal propagation between nodes. We can also use other signal propagation methods to model the initial representation of nodes in a certain view, and then observe the effect of multiscale neighbors. During the experiment, for the multiscale neighbor graph, we adopted different graph neural network models, such as NGCF, GCN, LightGCN, and GAT, to learn various neighbor embedding and observe the robust effect of multiscale neighbor signals. In the experiment, the hyperparameters $\lambda$, $\lambda_{Att}$, and $\lambda_{Int}$ are set to 0.01.

### 6. Experimental Results

In this subsection, we compare the proposed MSNAN model with the benchmark models in terms of MAE and RMSE metrics on three datasets. The experimental results are shown in Table 2. In Table 2, we have the following observations from the experimental results.

**Table 2.** MAE and RMSE results of different methods on three datasets.

| Method | ML-100kr | | BK-10C | | Douban | |
|---|---|---|---|---|---|---|
| | MAE | RMSE | MAE | RMSE | MAE | RMSE |
| NCF | 0.7457 | 0.9342 | 1.1611 | 1.5288 | 0.5781 | 0.7304 |
| NGCF | 0.7298 | 0.9195 | 1.1241 | 1.4776 | 0.5768 | 0.7271 |
| GCN | 0.7253 | 0.9178 | 1.1333 | <u>1.4772</u> | 0.5766 | 0.7259 |
| LightGCN | 0.7260 | 0.9182 | 1.1166 | 1.4794 | 0.5709 | 0.7213 |
| GAT | 0.7257 | 0.9186 | 1.1250 | 1.4813 | 0.5770 | 0.7238 |
| AFM | 0.7319 | 0.9238 | 1.1170 | 1.4786 | <u>0.5643</u> | <u>0.7136</u> |
| Wide&Deep | 0.7204 | 0.9152 | <u>1.1151</u> | 1.4807 | 0.5654 | 0.7141 |
| ACCM | <u>0.7145</u> | <u>0.9027</u> | 1.1983 | 1.5373 | 0.5789 | 0.7301 |
| TANP | 0.7881 | 0.9757 | 1.2081 | 1.5336 | 0.5767 | 0.7274 |
| MSNAN + NGCF | **0.6910** | 0.8842 | 1.1029 | 1.4586 | 0.5536 | 0.7069 |
| MSNAN + GCN | 0.7008 | 0.8890 | 1.1038 | 1.4598 | 0.5643 | 0.7091 |
| MSNAN + LightGCN | 0.6949 | **0.8815** | **1.1003** | **1.4563** | **0.5517** | **0.7003** |
| MSNAN + GAT | 0.6973 | 0.8901 | 1.1047 | 1.4607 | 0.5617 | 0.7034 |

Through the comparison results, the performance of MSNAN shows excellent improvements on three datasets. For example, compared with the ACCM method of collaborative filtering, the MAE and RMSE values of the MSNAN + LightGCN method increase by 2.74%, 2.35%, 8.18%, 5.27%, 4.70%, and 4.08% on three datasets, respectively. In addition, the MAE and RMSE of the MSNAN + NGCF model have achievements of 3.29%, 2.05%, 1.09%, 1.26%, 1.90%, and 0.94% over the best baseline on three datasets, respectively. Meanwhile, the improvements of the MSNAN + LightGCN model to the best baseline are 2.74% and 2.35% for MAE and RMSE on the ML-100kr dataset, 1.33% and 1.41% on the BK-10C dataset, and 2.23% and 1.86% on the Douban dataset, respectively. This shows that the proposed MSNAN model can better learn the embedding representation of users and items and effectively improve the accuracy of rating prediction, which verifies the significance of collaborative semantics of multiscale neighbors.

Compared with these graph neural network baselines, the proposed MSNAN based on multiscale neighbors achieves competitive improvements on all datasets. For example, on the ML-100kr dataset, the MSNAN + NGCF, MSNAN + GCN, MSNAN + LightGCN, and MSNAN + GAT methods improve by 5.32%, 3.83%; 3.38%, 3.14%; 4.28%, 4.00%; and 3.91%, 3.10% over the NGCF, GCN, LightGCN, and GAT baselines on MAE and RMSE metrics. In addition, on the BK-10C dataset, the MSNAN + NGCF, MSNAN + GCN, MSNAN + LightGCN, and MSNAN + GAT methods also improve MAE and RMSE values by 1.89%, 1.29%; 2.60%, 1.18%; 1.46%, 1.56%; and 1.80%, 1.39% over the corresponding graph model baselines, respectively. This is because the node embedding of MSNAN combines collaborative semantics of multiscale neighbors, which purifies the important information of similar neighbors in rating prediction. However, for different graph neural network models, the representation quality of nodes is different, and the superposition effect of MSNAN is also different. As is known, LightGCN simplifies the transformation matrix and activation function, which obtains the highest embedding quality of node representation, and can also improve the performance of this method. The performance of the general GCN method is relatively backward. In addition, compared with ACCM and AFM methods, the MSNAN approach can achieve a smaller error in the rating prediction scenario, which indicates that the proposed model can better learn the embedding representation of users and items with the collaborative signal of multiscale neighbors.

## 7. Discussion and Analysis

### 7.1. Sparsity Analysis

Sparse recommendation is a challenging problem in recommender systems. The sparsity problem of the user–item interaction matrix makes it difficult to learn the semantic representation of users and items, and reduces the accuracy of rating prediction results. In order to observe the robust advantage of multiscale neighbor signals for sparse recommendation, we conduct the comparative experiments by randomly shielding the rating labels and setting different sparsity proportions for three datasets. In the experiment, we compare the performance of proposed MSNAN model with several graph neural network methods on MAE and RMSE metrics. The experimental results are shown in Figures 2–7.

In the above figures, with the increase in data sparsity, the MAE and RMSE values of all methods show an increasing trend, which indicates that the error of rating prediction is gradually increasing. This demonstrates that high data sparsity can affect the quality of embedding representation of users and items and reduces the accuracy of recommendation results. In addition, with the improvement in data sparsity, the MAE and RMSE values of the graph neural network methods fluctuate. This is because the benchmark models of graph neural networks only consider one type of neighbor information of the user–item interaction matrix. Insufficient interaction records lead to the difficulties of higher-order information propagation of single-type neighbors, which limits the stability of node embedding representation.

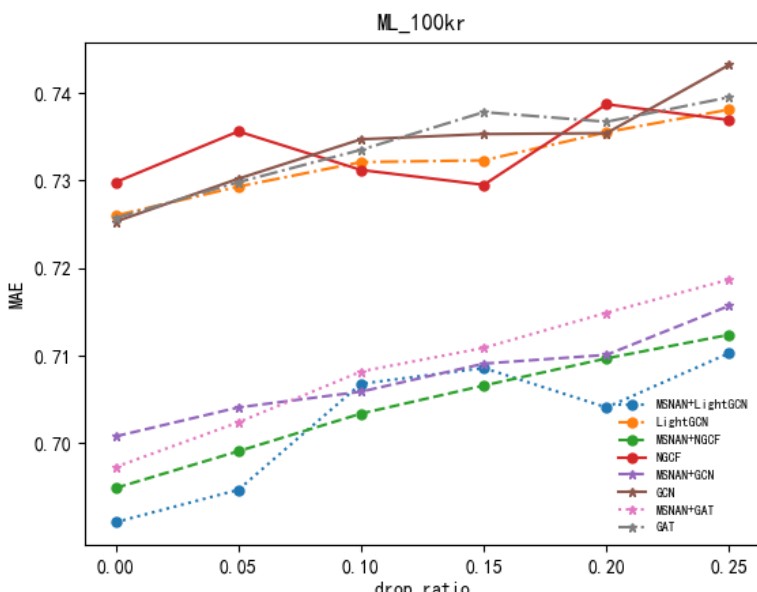

**Figure 2.** MAE results of different methods on the ML-100kr dataset with different sparsity proportions.

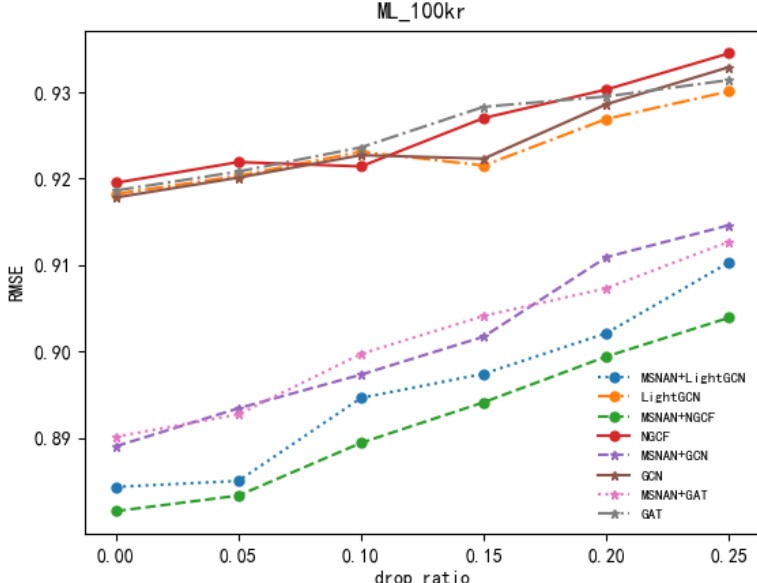

**Figure 3.** RMSE results of different methods on the ML-100kr dataset with different sparsity proportions.

Compared with graph neural network methods, the proposed MSNAN model consistently yields the best performance for MAE and RMSE on three datasets to adapt the sparsity ratio of different scales. For different graph neural network models, the result of blending the MSNAN model is more generalized than the original model. As shown in Figures 4–7, when there is a low drop ratio, LightGCN can obtain better performance compared with other graph neural network methods due to its better node representation quality. This is because LightGCN itself effectively learns the embedding representation of nodes by simplifying the nonlinear structure and reducing complexity. In large-scale e-commerce platforms, the sparsity of user–item interactions can be high. It is difficult to find collaborative neighbors based on the similarity of interaction behaviors. Although the user–item matrix loses a part of interaction records, the proposed model fuses various neighbor information from attribute views and rating views, which fully learns the representations of users and items. According to the attributes of users or items, we can find neighbors of different scales and conduct collaborative filtering recommendation.

Moreover, the model takes advantage of high-quality collaborative signals from multiscale neighbors for improving the quality of embedding representation, which is suitable for the practice of large-scale e-commerce and alleviates the performance impact of data sparsity to some extent.

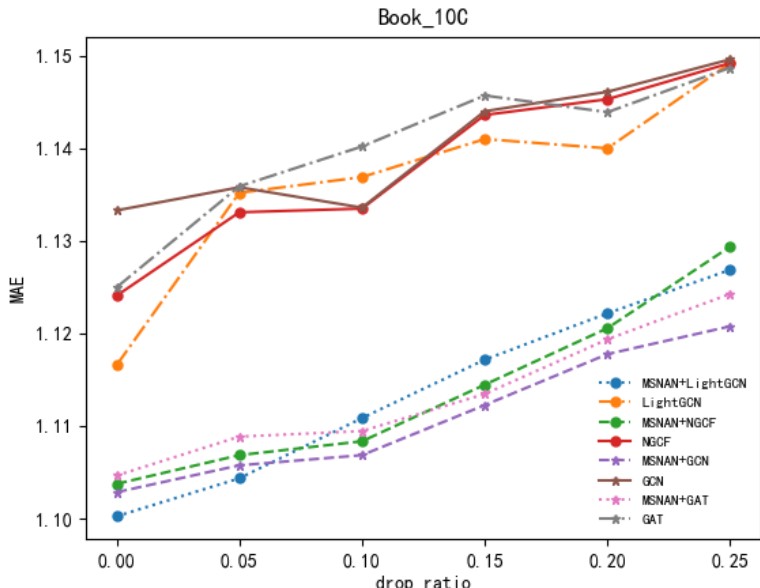

**Figure 4.** MAE results of different methods on the BK-10C dataset with different sparsity proportions.

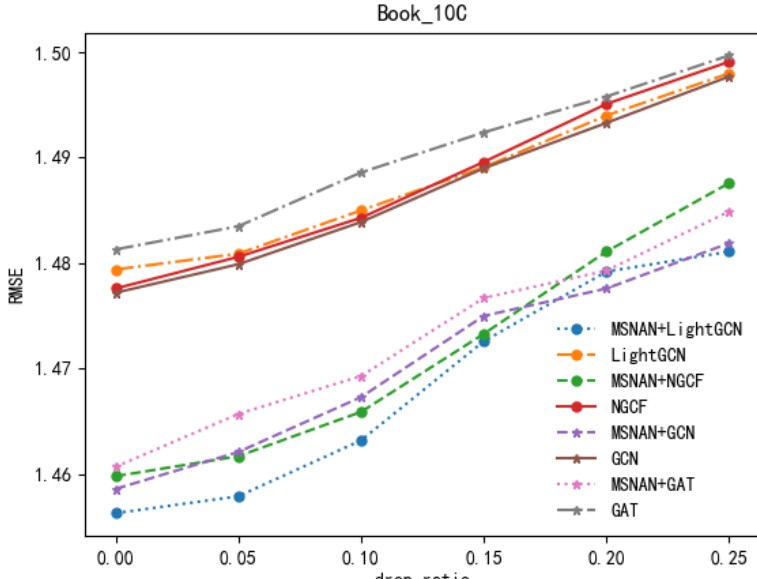

**Figure 5.** RMSE results of different methods on the BK-10C dataset with different sparsity proportions.

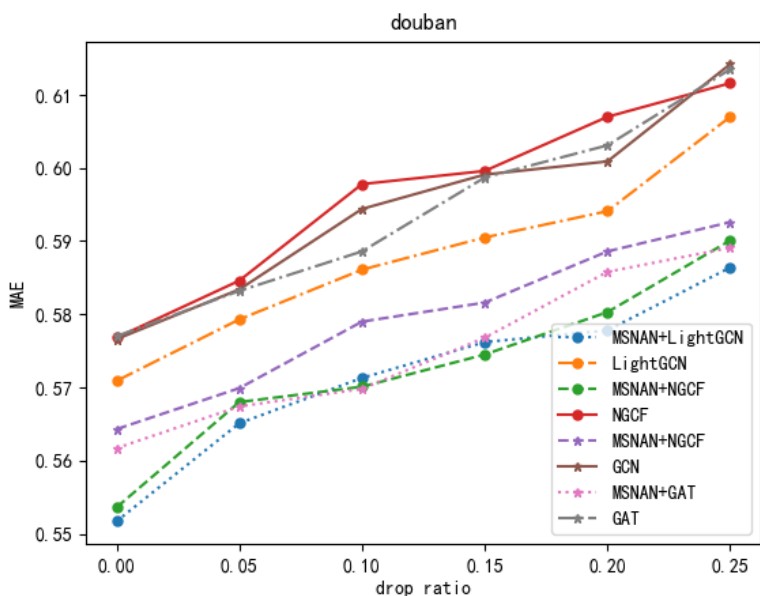

**Figure 6.** MAE results of different methods on the Douban dataset with different sparsity proportions.

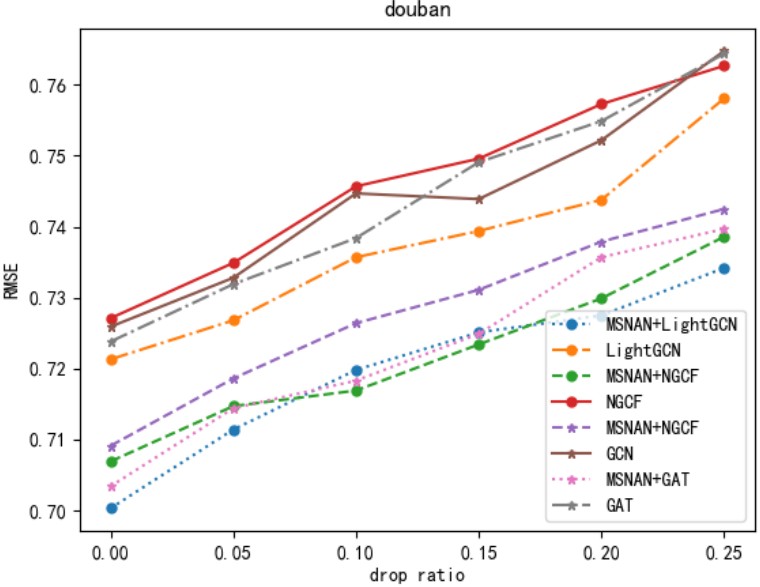

**Figure 7.** RMSE results of different methods on the Douban dataset with different sparsity proportions.

*7.2. Impact of Neighbor Embedding*

For the multiscale neighbor embedding, the neighbor embedding of different views contributes differently, which affects the result of rating prediction. To observe the roles of different types of neighbor embedding, Figures 8–10 give the performance of methods by removing different attribute-view neighbor embedding. In Figure 8, for the ML-100kr dataset, the users have three types of neighbor embedding. We can see that the MAE and RMSE of three att-view neighbor embedding change greatly compared with the other two types, which states that three att-view neighbor embedding plays an important role in predicting the interest rating of the target user. That is, through the neighbor interaction graphs with multiple similar attributes, the multiscale neighbor embedding achieves a stronger ability to express the preference of target user, which also demonstrates that neighbors with fine-grained interests play a greater role in collaborative recommendation. Similarly, for the BK-10C dataset, after removing the four att-view neighbor embedding, the MAE and RMSE values change the most, which indicates that four att-view neighbor

embedding makes the greatest contribution on modeling the preference of the user. For the Douban dataset, two att-view neighbor embedding is important. These results indicate that multiscale neighbors on different views have a collaborative effect on the user's preference decisions.

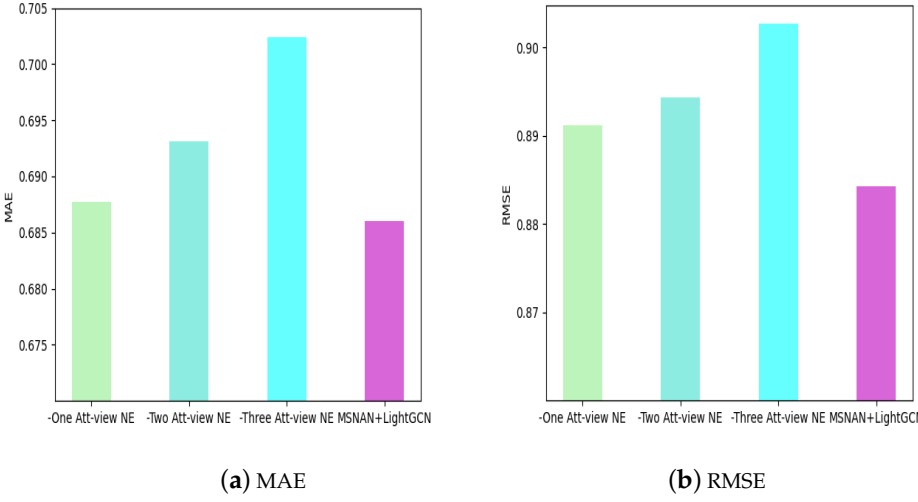

(**a**) MAE　　　　　　　　　　　　　　(**b**) RMSE

**Figure 8.** Ablation study of multiscale neighbor embedding on the ML-100kr dataset.

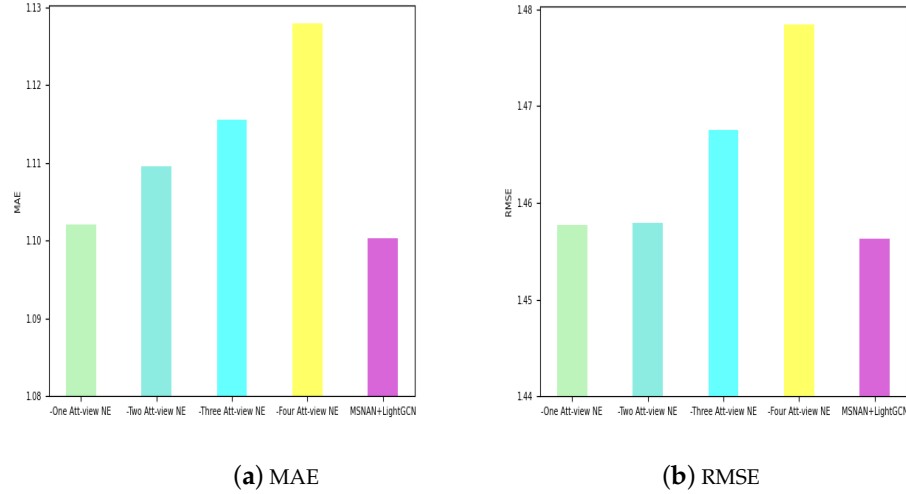

(**a**) MAE　　　　　　　　　　　　　　(**b**) RMSE

**Figure 9.** Ablation study of multiscale neighbor embedding on the BK-10C dataset.

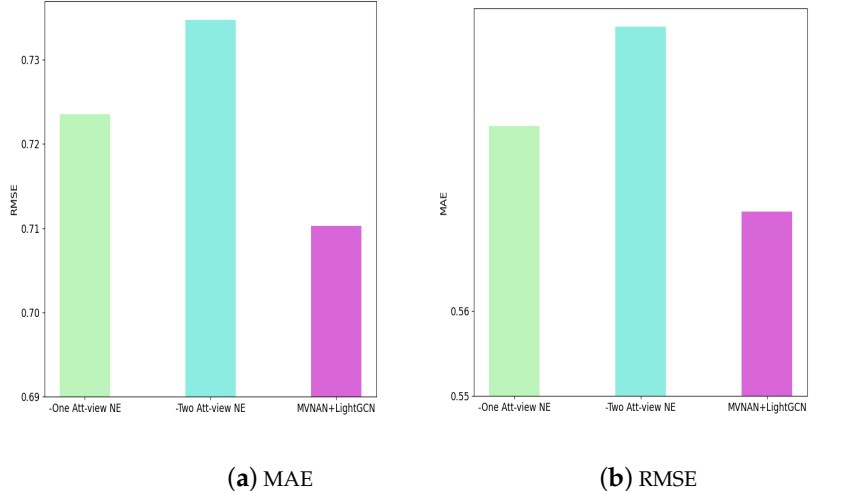

(**a**) MAE　　　　　　　　　　　　　　(**b**) RMSE

**Figure 10.** Ablation study of multiscale neighbor embedding on the Douban dataset.

### 7.3. Visual Explanation Study

In order to further observe the influence of multiscale neighbor embedding on user decision-making, Figures 11–13 report the attention weight of attribute-view neighbor embedding and interaction-view neighbor embedding of 10 users for three datasets. In Figure 11a, the weight of three att-view is the largest, which indicates that similar neighbors with multiple attributes can have a major contribution on capturing collaborative signals. One att-view neighbor embedding can also capture coarse-grained semantic interests for collaborative filtering. This important insight allows us to select neighbors for users and items with different numbers of attributes, and capture rich signals for collaborative filtering recommendation tasks. In addition, in Figure 11, the user's final embedding representation is most affected by the four att-view embedding and eight sco-view embedding. By observing the dataset of BK-10Core, we find that most users are used to evaluating the items with high score values, which leads to a large contribution of the high-score-view embedding. Meanwhile, the rating neighbor matrix is relatively sparse, which makes the difference between the weight values of various types of high-score-view embeddings small.

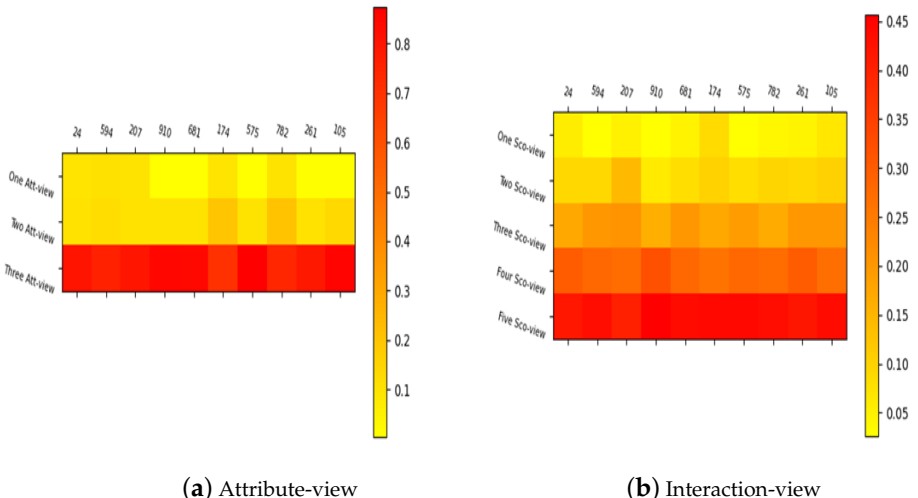

(**a**) Attribute-view        (**b**) Interaction-view

**Figure 11.** Attention weight of multiscale neighbor embedding for 10 users on the ML-100kr dataset.

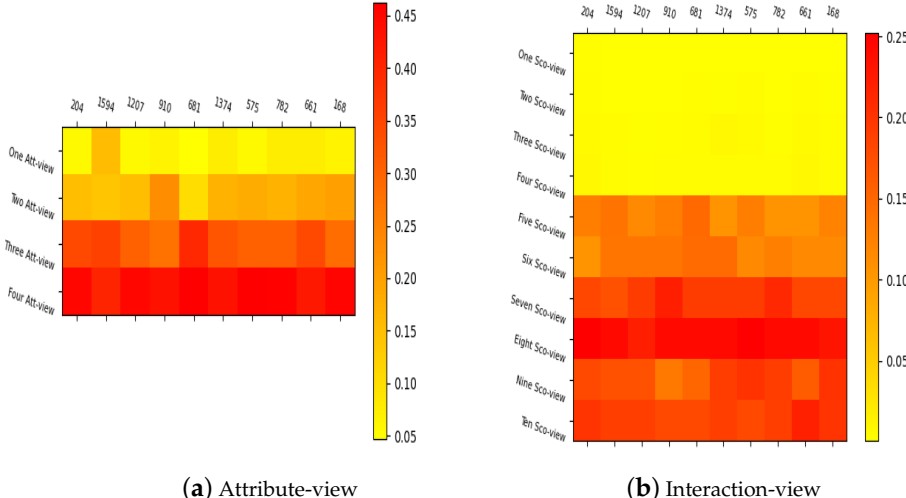

(**a**) Attribute-view        (**b**) Interaction-view

**Figure 12.** Attention weight of multiscale neighbor embedding for 10 users on the BK-10C dataset.

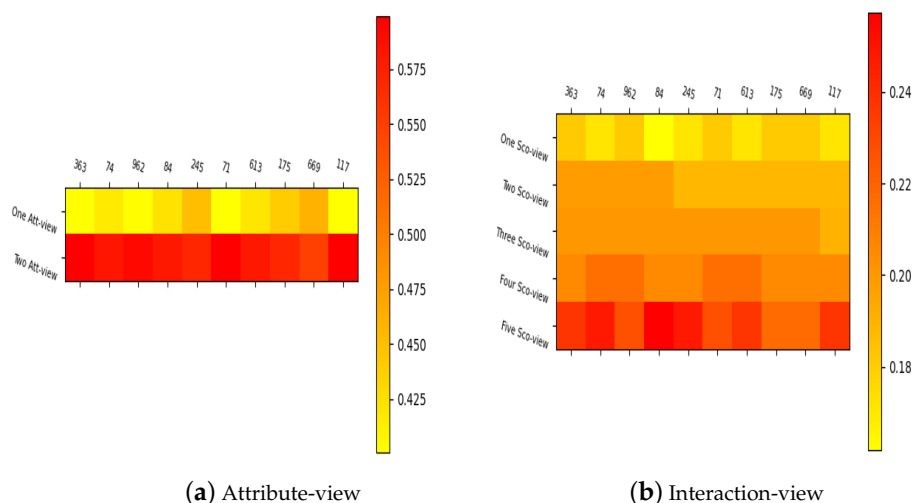

(**a**) Attribute-view         (**b**) Interaction-view

**Figure 13.** Attention weight of multiscale neighbor embedding for 10 users on the Douban dataset.

## 8. Conclusions

In this paper, we propose a multiscale neighbor-aware attention network for collaborative filtering recommendation. The proposed strategy fuses the global semantics of various types of neighbors and important local embedding of multiscale neighbors. Multiple attribute-view neighbors and interaction-view neighbors provide collaborative signals to predict the user's rating of items. Experiments verify the effectiveness of collaborative contributions of multiscale neighbors for learning user and item representation. The key finding is that the combination of multiscale attribute neighbors and interactive neighbors can improve the accuracy of recommendation, and alleviate the poor performance of recommendation in the case of sparse data. The disadvantage is that the computation of multiscale neighbors requires different graph structures to learn the representation of nodes. The platform can construct the graph structure offline and calculate multiscale neighbors, which can reduce the online resource requirements. However, in the e-commerce scenario, the proposed method can realize targeted personalized recommendation according to different attribute neighbors of users. In particular, for some cold-start users who do not have interactive behaviors, the method can select neighbors with similar attributes for the target user according to one's social attribute set, and then conduct collaborative filtering recommendation. Some products and services can be recommended based on similar attribute preferences of neighbors for target users. In addition, by combining with the behavioral preferences of group users, we can make rating predictions and recommend popular products for target users.

In future work, by investigating the semantic difference of various attribute and interactive behavior views, we can focus on a consistency study of node representations on different behavior views and improve the accuracy and interpretability of the recommendation system. In addition, heterogeneous types of semantic information from different types of user behaviors such as evaluation, clicking, and buying can describe the ordered semantic interests of the user. We will distinguish the types of multiple interaction behaviors to learn heterogeneous semantic representations and model the sequential relations between different behaviors.

**Author Contributions:** Methodology, J.Z., F.C. and Q.C.; Software, T.J.; Investigation, J.Z. and Y.L.; Writing—original draft, J.Z.; Writing—review & editing, Y.K. and Q.C. All authors have read and agreed to the published version of the manuscript.

**Funding:** This work was partially supported by the National Natural Science Foundation of China (nos. 62272286, 62072291), the Natural Science Foundation of Shanxi Province (nos. 20210302123468, 202203021221021, 202203021221001).

**Data Availability Statement:** Data used in this manuscript consist of publicly available standard benchmark datasets.

**Conflicts of Interest:** The authors declare no conflict of interest.

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
