# Peer review of "A Multiscale Neighbor-Aware Attention Network for Collaborative Filtering"

_electronics, doi:10.3390/electronics12204372_

Round 1

Reviewer 1 Report

I would like to thank the authors for the opportunity to read their paper. They proposed a multiscale neighbor-aware attention network for collaborative filtering.

The paper is well conducted, the methodology correctly applied, and the literature review is comprehensive.

The references are appropriate. There are 40+ references in this paper, the bibliography is recent and adequate for the research.

Overall, it is a study with definite scientific value, which makes significant contributions to the field of research.

However, from my point of view, the authors must develop the conclusion section.

Reviewer 2 Report

This paper proposed a multi-scale neighbor-aware attention network for collaborative filtering in recommender system. The proposed method improved the accuracy and solved sparse data scenario. My concerns are listed in the following.

  1. Why do you design the multi-scale neighbor-aware  attention network in Fig. 1? What is the function of each part in that framework?
  2. Can you give some analysis about the different impact of NGCF/GCN/LightGCN/GAT in conjunction with MSNAN? (Table 2)
  3. Why MSNAN has the best performance in conjunction with NGCF? (Table 2)
  4. From Fig. 4 - 7, I find that MSNAN+LightGCN has the best performance at low drop ratio. Can you give some analysis about this phenomenon?

Reviewer 3 Report

My comments:
1. The topic of this paper is interesting and innovative and it will contribute in related research field.

2. The sections of “3. Framework of multiscale neighbor-aware attention network” and “4. Methodology” are well-written.

3. I suggest authors to separate the section “Experimental results and analysis” into two sections-- “Results” and “Discussion”.

4. The section of “7. Conclusion” must be reinforced more. For example, the contributions to academic research as well as theoretical implications and research limitations.

Minor editing of English language required

Reviewer 4 Report

Complexity: While the proposed MSNAN approach is innovative, its complexity may present challenges in terms of implementation and computational resources required. Integrating multiscale neighbors and attention networks might result in higher processing demands, making it less suitable for resource-constrained environments.

Limited Comparison: While the paper compares the proposed MSNAN method with traditional methods and demonstrates improvements in MAE and RMSE, a more extensive comparison with a wider range of state-of-the-art collaborative filtering techniques would provide a clearer understanding of its competitive advantages.

Real-World Scalability: The paper's effectiveness is demonstrated on datasets, but its real-world scalability and performance in large-scale recommender systems with millions of users and items remain unaddressed. Future research should explore the practicality and efficiency of MSNAN in such contexts.

Other: the paper did not consider the cold start problem. 
